Algae from Aiptasia egesta are robust representations of Symbiodiniaceae in the free-living state

Maruyama Shumpei shumpeim@gmail.com 1
Unsworth Julia R. 2
Sawiccy Valeri 1
Students of Oregon State University’s Z362 Spring 2021 1
Weis Virginia M. 1
1 Department of Integrative Biology, Oregon State University , Corvallis , OR , United States of America
2 Department of Biology, Lewis and Clark College , Portland , OR , United States of America
Pochon Xavier
Electronic publication date: 2022 Jul 29
Publication date: 2022
Volume: 10
Electronic Location ID: e13796
Received 2022 Apr 21; Accepted 2022 Jul 6
Copyright: ©2022 Maruyama et al.
Copyright year: 2022
Copyright holder: Maruyama et al.
License: This is an open access article distributed under the terms of the Creative Commons Attribution License, which permits unrestricted use, distribution, reproduction and adaptation in any medium and for any purpose provided that it is properly attributed. For attribution, the original author(s), title, publication source (PeerJ) and either DOI or URL of the article must be cited.
License URL: https://creativecommons.org/licenses/by/4.0/

Keywords: Symbiosis, Aiptasia, Symbiodiniaceae, Algae, Coral, Cnidaria

Funding: National Science Foundation IOS 1529059 IOS:EDGE 1645164 This work was supported by grants IOS 1529059 and IOS:EDGE 1645164 from the National Science Foundation to Virginia M Weis. The funders had no role in study design, data collection and analysis, decision to publish, or preparation of the manuscript.

==============================
Many cnidarians rely on their dinoflagellate partners from the family Symbiodiniaceae for their ecological success. Symbiotic species of Symbiodiniaceae have two distinct life stages: inside the host, in hospite, and outside the host, ex hospite. Several aspects of cnidarian-algal symbiosis can be understood by comparing these two life stages. Most commonly, algae in culture are used in comparative studies to represent the ex hospite life stage, however, nutrition becomes a confounding variable for this comparison because algal culture media is nutrient rich, while algae in hospite are sampled from hosts maintained in oligotrophic seawater. In contrast to cultured algae, expelled algae may be a more robust representation of the ex hospite state, as the host and expelled algae are in the same seawater environment, removing differences in culture media as a confounding variable. Here, we studied the physiology of algae released from the sea anemone Exaiptasia diaphana (commonly called Aiptasia), a model system for the study of coral-algal symbiosis. In Aiptasia, algae are released in distinct pellets, referred to as egesta, and we explored its potential as an experimental system to represent Symbiodiniaceae in the ex hospite state. Observation under confocal and differential interference contrast microscopy revealed that egesta contained discharged nematocysts, host tissue, and were populated by a diversity of microbes, including protists and cyanobacteria. Further experiments revealed that egesta were released at night. In addition, algae in egesta had a higher mitotic index than algae in hospite, were photosynthetically viable for at least 48 hrs after expulsion, and could competently establish symbiosis with aposymbiotic Aiptasia. We then studied the gene expression of nutrient-related genes and studied their expression using qPCR. From the genes tested, we found that algae from egesta closely mirrored gene expression profiles of algae in hospite and were dissimilar to those of cultured algae, suggesting that algae from egesta are in a nutritional environment that is similar to their in hospite counterparts. Altogether, evidence is provided that algae from Aiptasia egesta are a robust representation of Symbiodiniaceae in the ex hospite state and their use in experiments can improve our understanding of cnidarian-algal symbiosis.

Introduction

The family Symbiodiniaceae is a diverse group of dinoflagellates, many of which form symbioses with a variety of cnidarian hosts, including corals and sea anemones. In this symbiosis, the algal symbiont is housed within host gastrodermal cells where they provide photosynthate to the host and in return, the host provides inorganic nutrients, protection and a high light environment (Davy, Allemand & Weis, 2012; Marcelino et al., 2013). The two partners must work cooperatively to maintain and regulate symbiosis. For example, the host utilizes carbonic anhydrase and bicarbonate transporters to supply inorganic carbon to the symbiont to maximize photosynthesis (Tansik, Fitt & Hopkinson, 2017; Koch, Verde & Weis, 2020). Reciprocally, the symbiont may be modulating the host’s immune system to prevent immune destruction or expulsion by the host (Detournay et al., 2012; Mansfield et al., 2017; Jacobovitz et al., 2021; Jinkerson et al., 2022). However, our understanding of the cellular mechanisms involved in maintaining symbiosis remains limited.

Studies often compare the biology of Symbiodiniaceae between its two life stages: inside the host, in hospite and outside the host, ex hospite. In this comparison, algae that are freshly isolated from the host are used to represent algae in hospite, while cultured algae are often used to represent algae ex hospite. Studies have found that compared to algae in culture, algae in hospite have lower cell division rates (Smith & Muscatine, 1999; Tivey, Parkinson & Weis, 2020) and differential expression of genes that are tightly linked to nutrient limitation (Maor-Landaw, Van Oppen & McFadden, 2020; Xiang et al., 2020; Cui et al., 2022). Based on these data, it is commonly hypothesized that the host limits nutrient transfer to the symbiont as a mechanism to maintain a stable population of symbionts (Falkowski et al., 1993; Smith & Muscatine, 1999; Xiang et al., 2020; Tivey, Parkinson & Weis, 2020; Cui et al., 2022). This comparison, however, introduces nutrition as a confounding variable because while algae in hospite are sampled from hosts maintained in oligotrophic seawater, algae ex hospite are sampled from cultures grown in nutrient-rich media. The nutrient-depleted phenotype found in algae in hospite could therefore not be caused by host-inhibitory mechanisms and instead may simply reflect the low nutrient availability in the water column surrounding hosts (for in-depth discussion, see Maruyama & Weis, 2021). In fact, studies have found that supplementing nutrients to the host, by feeding or by the addition of inorganic nutrients to the water column, will return the algae to a nutrient-enriched phenotype as measured by cell division rates or gene expression profiles, suggesting that the host readily supplies nutrients to their symbionts when available (Hoegh-Guldberg & Smith, 1989; Stambler et al., 1991; Falkowski et al., 1993; Hoegh-Guldberg, 1994; Muller-Parker et al., 1994; Smith & Muscatine, 1999; Ferrier-Pagès et al., 2001; Houlbrèque et al., 2004; Rosset, D’Angelo & Wiedenmann, 2015; Xiang et al., 2020; Tivey, Parkinson & Weis, 2020; Cui et al., 2022). Further work is necessary to tease apart the effects of symbiosis and nutrient availability on algal phenotypes to determine whether the host is limiting nutrient transfer to its symbionts.

Multiple transcriptomic studies have found that nutrient-related genes are differentially expressed between algae in hospite and those in culture (Maor-Landaw, Van Oppen & McFadden, 2020; Xiang et al., 2020; Cui et al., 2022). Examples include ammonium transporter AMT, nitrate transporter NRT, purine nucleoside permease NUP, and nitrate reductase NR that are upregulated and glutamine synthetase GS that is downregulated in algae in hospite compared to those in culture. This gene expression pattern is indicative of nutrient limitation in Symbiodiniaceae and other marine algae (Hildebrand, 2005; Kang et al., 2007; Xiang et al., 2020). Some differentially expressed genes, however, are probably not linked to nutrition, such as the sugar transporter SWEET1, that is hypothesized to play a role in transferring photosynthate to the host. Sampling a more robust representation of Symbiodiniaceae in the ex hospite state that controls for nutrient availability could confirm whether these differentially regulated genes are reflective of nutrition, the symbiotic state, or both.

In cnidarian-algal symbiosis, the host regularly expels viable algae as a homeostatic mechanism to maintain a stable population of symbionts (Steele, 1975; Hoegh-Guldberg, McCloskey & Muscatine, 1987; Hill & Ralph, 2007; Thornhill et al., 2017). In this study, we hypothesized that sampling expelled algae as a representation of the ex hospite state in Symbiodiniaceae for comparison with their in hospite counterparts could remove nutrition as a confounding variable, as both the host and the expelled algae are in the same oligotrophic seawater environment. Therefore, physiological and/or gene expression differences found between algae in hospite and expelled algae in this comparison would be modulated by the host, and not be masked by the high-nutrient state of cultured algae as in the traditional comparison. In the sea anemone Exaiptasia diaphana (commonly called Aiptasia), a model system for the study of cnidarian-algal symbiosis, algae are released in distinct, brown pellets, herein referred to as egesta. The brown egesta primarily consist of algae, and are released separately from pellets containing digested food (Steele, 1975). The discrete nature of egesta, with their high abundance of algae, made them a promising experimental system for sampling purposes.

To begin the practice of sampling expelled algae in symbiosis studies, we first explored the basic biology of algae from Aiptasia egesta and characterized egesta appearance, timing of egesta expulsion, and characteristics of the algae including mitotic index, photophysiology, and ability to initiate symbiosis with new hosts. Then, we studied the gene expression of several nutrient-related genes to determine the nutrient status of algae from egesta. Finally, we discuss several practical considerations of the use of algae from Aiptasia egesta as an experimental model system to represent Symbiodiniaceae in the ex hospite state.

Materials & Methods

Animal and algal culture maintenance

Aiptasia (Exaiptasia diaphana, clone ID: H2) symbiotic with Breviolum minutum (culture ID: SSB01) and SSB01 cultures were used in this study. Animals were maintained in Coralife Instant Ocean (Tempe, AZ, USA) artificial seawater (ASW), and algae were maintained in F/2 media made in filtered artificial seawater (FSW). Animals and algae were housed at 25 °C in a Percival AL-41L4 incubator (Perry, IA, USA) with Zoo Med Laboratories (San Luis Obispo, CA, USA) 10,000 k fluorescent lamps set to 40 µmol photons/m2/s using shadecloth on a 12 h:12 h light:dark cycle. Aposymbiotic anemones were originally bleached using menthol (Matthews et al., 2016) and maintained symbiont-free by incubation in the dark for several months before use. Unless otherwise stated, animals were fed three times a week ad libitum with freshly hatched brine shrimp and the water was changed three times a week with ASW.

Confocal imagery and light microscopy

Live egesta samples were imaged on a Zeiss LSM 780 NLO confocal microscope system (Zeiss, Oberkochen, Germany). Images were taken on three separate channels, with excitation from a 405 nm Diode, Argon (488), and 633 nm HeNe. Emission was detected between 410–470 nm, 491–572 nm, and 647–722 nm, respectively. Simultaneously, using laser illumination, a transmitted light detector (T-PMT) acquired transmitted light images in bright field. Confocal z-stack images were processed for pseudocoloring and merged to create a maximum intensity projection using ZEN Black (Carl Zeiss AG) and ImageJ software. Separately, differential interference contrast (DIC) images were obtained using an Olympus Vanox-T AH2 microscope to image nematocysts.

Time-lapse of egesta release

Sixteen to twenty Aiptasia were transferred to twenty cm diameter glass Carolina Biological (Burlington, NC, USA) culture dishes with one L of ASW. Animals were allowed to settle for at least 24 hrs prior to time-lapse photography. Immediately before imaging began, the culture dish was cleaned with a cotton swab and the water was changed. For time-lapse imaging, a Canon 5D Mark II DSLR with external flash was set to photograph the entire dish hourly for 24 hrs. To control for any effect of cleaning on timing of egesta release, two time-lapses (Experiment 1 & 2) began at 16:30 and two other time-lapses (Experiment 3 & 4) began at 22:30. Following their original light schedule, lights were set to come on at 7:00 and off at 19:00. Animals were not fed during the experiment. Each image was analyzed with ImageJ using the Cell Counter plugin, and each dark particle in an image were counted as egesta. The number of egesta in each time point was subtracted from the total number of egesta in the previous time point to calculate the quantity of egesta released in the hour. The quantity of egesta was then normalized to anemone number per container to obtain the number of egesta per anemone at each time point.

Mitotic index

Thirty-two Aiptasia were housed individually in two mL FSW in 24-well plates and allowed to settle for 24 hrs. After settlement, each well was cleaned with a cotton swab and the water was changed. Egesta were collected after an additional 24 hrs. Of the 32 original anemones, eight released egesta large enough to be imaged. The eight anemones that released usable egesta were homogenized with a microcentrifuge pestle in 300 µL FSW, spun down at 800× g for five min, and resuspended in twenty µL FSW. Resuspended algae and egesta were then imaged under DIC light microscopy.

Images were analyzed using ImageJ with the Cell Counter plugin by students enrolled in Oregon State University’s Spring 2021 Z362 Invertebrate Biology laboratory. Following the method of Baghdasarian & Muscatine (2000), individual cells in each image were scored based on presence of a cell wall division plate, and at least 700 cells were counted per sample. Students were tasked to save their cell counter files, and their counts were manually verified by loading the counter files onto the images. The mitotic index was calculated as the percent of cells in the entire population that were undergoing division and paired t-tests were performed for statistical analysis.

Algal photophysiology and cell viability

Egesta were collected from Aiptasia less than 24 hrs after last cleaning, to ensure collection of freshly expelled egesta. All collected egesta were pooled together and pelleted at 800× g for five min, the supernatant was removed and the cells resuspended in one mL FSW. The pooled sample was divided into two 500 µL aliquots. One was kept intact, while the other was homogenized using a motorized microcentrifuge pestle. From the homogenized sample, twenty µL was used to quantify cell number using a Thermo Fisher Scientific (Waltham, MA, USA) Countess II automated cell counter. Homogenized and intact egesta samples were then dispensed into twelve well plates in 95 µL (20,000 cells per sample, n = 5) and FSW was added up to two mL. Cultured algae (SSB01) was also collected, pelleted at 800× g for five min and resuspended in FSW. Five culture samples of 20,000 cells were then dispensed in twelve well plates and FSW added up to two mL. All plates were then returned to their original incubator.

Maximum quantum yield of photosystem II (Fv/Fm) was obtained using a Light Induced Fluorescence Transient Fast Repetition Rate fluorometer (LIFT-FRRf; Soliense Inc, Santa Cruz, CA). Fv/Fm was measured once a day at 13:00 following dark acclimation for 30 min to ensure measurement of maximum quantum yield. To measure Fv/Fm, excitation was delivered at 475 nm wavelength in four phases and fluorescence detected at 685 nm. The first phase was a saturating sequence of 100 flashlets, each lasting 0.7 µs with a 1.5 µs gap. The second phase was a relaxation phase of 80 flashlets beginning with a gap of twenty µs, increasing exponentially until the end of the sequence. The third phase was a sequence of 1,600 flashlets lasting two µs each with a 40 µs gap. The final relaxation phase was identical to the second phase. Data were fitted using LIFT software with the three-component exponential model (SEQ_3).

To test the ratio of live and dead cells, we used an Evans Blue cell viability dye (Morera & Villanueva, 2009). Approximately ten pellets of egesta less than 24 hrs old were collected and homogenized together using a motorized microcentrifuge pestle in 500 µL of ASW. Cells were then pelleted at 800× g for five min and the pellet resuspended in 100 µL of FSW. Cells were then stained with 20 µL of Evans Blue dye and incubated for ten min. The sample was imaged under light microscopy and algal cells that internalized the blue dye were counted as dead while cells that were not stained were counted as alive.

Inoculation of aiptasia with algae

Aposymbiotic anemones (n = 9) were plated into 24 well plates in two mL FSW. Plates were placed in their original incubators set on a 12 h:12 h light:dark cycle for 72 h. After incubation in the light, animals were manually checked under fluorescence microscopy to verify aposymbiotic status.

Egesta were collected from Aiptasia less than 24 hrs after last cleaning. Samples of both egesta and algae in culture were pelleted at 800× g for five min and resuspended in 300 µL FSW. To obtain freshly isolated symbionts, one symbiotic anemone was homogenized using a motorized microcentrifuge pestle in 300 µL of FSW. Egesta and cultured algae were also homogenized using a motorized microcentrifuge pestle. All samples were then washed twice by centrifugation at 800× g for five min and resuspension in two mL FSW. Algal samples were counted using a Countess II automated cell counter, and samples were diluted to 1 × 106 algal cells/mL in FSW.

To inoculate aposymbiotic Aiptasia, 1 × 105 algal cells in a 100 µL volume were gently pipetted over anemone mouths. Immediately after the addition of algae, twenty µL of brine shrimp extract was pipetted to anemones to induce a feeding response. Negative controls were fed the brine shrimp extract with no algae added. Anemones were allowed to take up algae for 24 hrs. The water was then changed, and anemones were moved to clean wells of a 24 well plate in two mL FSW. After an additional 24 hrs, the anemones were imaged under fluorescence microscopy. Briefly, anemones were relaxed in 0.18 M MgCl2 dissolved in FSW, and images of three tentacles per anemone were taken in several focal depths under brightfield and under the red Filter Set 15 (Carl Zeiss) to visualize algal cell auto-fluorescence using a Zeiss Axio Observer inverted microscope. Each manually constructed z-stack was merged into a single image using Adobe Photoshop. Symbiont density was determined from each image using ImageJ as the number of fluorescent algal cells normalized to two-dimensional tentacle area.

Primer design

Target genes were selected based on their function and differential expression between algae in culture and in hospite from Xiang et al. (2020). Primer and transcript sequences were obtained from Xiang et al. (2015) and Xiang et al. (2020) for an ammonium transporter (AMT, transcript ID: s6_38207), a nitrate transporter (NRT, s6_422), purine nucleoside permease (NUP, s6_27864), and nitrate reductase (NR, s6_34). New primers were designed for glutamine synthetase (GS, s6_5551), photosystem II protein D1 (psbA, s6_1009), and a sugar transporter (SWEET1, s6_35311). We chose to study the expression of psbA because its expression was consistent between algae in culture and in hospite (Xiang et al., 2020). Cyclophilin was chosen as an algal housekeeping gene (Rosic et al., 2011) and primers were designed to be specific for B. minutum and not amplify Aiptasia cyclophilin. Primer pairs were verified for amplification efficiency of at least 90% by assessing the slope of standard curves generated from quantitative PCR reactions of serial dilutions of template cDNA. Primers sequences are available in the supplementary data.

Quantitative PCR

To quantify gene expression, 54 anemones were placed in three separate containers with 500 mL ASW (eighteen anemones per container). The average oral disk diameter of the anemones was 5.4 mm (data not shown). Anemones were fed every other day ad libitum. Egesta containing algae were collected 24 hrs after feeding, and the container cleaned after egesta were collected. We did not sample egesta containing digested food and they are easy to distinguish due to the pink color of the cartenoids from digested Artemia sp. nauplii. To obtain enough material for gene expression studies (approximately 1 × 106 cells), egesta were collected a total of six times from each container. Immediately after collection, egesta were processed for RNA extraction as follows. First, egesta were homogenized in cold FSW with a motorized microcentrifuge pestle. To completely isolate algae from host cells, the sample was then further homogenized by passing the sample seven times through a three mL syringe fitted with a 23 gauge needle. Following protocols from Xiang et al. (2020), the homogenate was then loaded on a 50% isotonic Percoll solution made in FSW and spun for twenty min at 9,000× g. The supernatant containing host material was carefully removed, and the algae were resuspended in one mL cold FSW. The sample was then pelleted at 3,000× g for five min and the supernatant removed. The pellet was then frozen at −80 °C until further processing.

To obtain algae in hospite, four random anemones from each container were sampled immediately after the last collection of egesta and processed for RNA extraction following the same procedure as those for egesta, and the algal pellets frozen for at least ten min. Three independent, but identical cultures of SSB01 grown in F/2 culture media were made simultaneously and allowed to grow for three weeks. These cultures were also sampled and followed the same procedure for RNA processing and the algal pellets were frozen for at least ten min.

After the final sample of egesta was collected, the frozen samples from each repeated collection were pooled together corresponding to their respective Aiptasia containers. Each frozen freshly isolated, culture, and egesta sample was then resuspended in 300 µL Trizol. Sterilized glass beads were added to the sample, and the sample was homogenized with a bead beater for two min set at 4,000 RPM. The beads were then removed, the samples were centrifuged at 16,000× g for one min, and the supernatant containing RNA was extracted. The RNA was purified using a Zymo Direct-zol kit following the manufacturer’s instructions. RNA samples were further processed using a Turbo DNA-free kit to remove genomic DNA contaminants. Samples were then purified again using a New England Biolabs (Ipswich, MA, USA) Monarch RNA purification kit, followed by a One-Step PCR inhibitor removal kit to remove carbohydrate contaminants. The RNA samples were analyzed for purity and concentration using a Nanodrop.

All samples were then diluted to ten ng/µL and cDNA was synthesized from nine µL (90 ng total) of RNA in 30 µL reactions using the New England Biolabs Protoscript II cDNA synthesis kit with oligo d(T)23 VN primers following the manufacturer’s instructions. Quantitave PCR reactions were performed in twenty µL volumes using 1X Power SYBR Green PCR Mastermix, 0.5 µM of each primer pair, and one µL of template cDNA. A BioRad CFX96 Real-Time System was used to carry out reactions using a two-step amplification phase (95 °C/ten min, followed by 40 cycles of 95 °C/ten s and 58 °C/30 s) and then a melting-curve analysis performed to confirm the presence of single amplicons. No-reverse transcription, no-primer, and no-template controls were included as negative controls.

For each sample and target gene, ΔCt values were calculated by subtracting the Ct value of the target gene from the Ct value of the housekeeping gene (cyclophilin). Then, the ΔCt value was subtracted from the mean ΔCt value of corresponding genes from cultured algae samples to calculate relative expression ΔΔCt. Fold gene expression was then calculated as 2−ΔΔCt for each gene from each sample. Statistical analyses were conducted on ΔΔCt values for each gene by ANOVA and post-hoc Tukey’s HSD.

Results

Physical characteristics of egesta

Egesta observed under confocal microscopy revealed several autofluorescent features (Fig. 1). Chlorophyll autofluorescence was readily apparent from B. minutum under 633 nm excitation. Under 488 nm excitation, autofluorescence of putative pyrenoids from B. minutum, phycobillins from cyanobacteria, and host tissue were detected (Fig. 1A). A signal was also detected under 405 nm excitation within egesta (Fig. 1A), and we hypothesize that its origin comes from dead remains of an unidentified ciliate protist that autofluoresced under the same excitation wavelength (Fig. 1C). Differential interference contrast microscopy revealed that the surface of egesta contained discharged nematocysts (Fig. 1D) and the remains of nematocyst capsules (Fig. 1E). Egesta were also colonized by several kinds of unidentified protists and microbes (data not shown). None of the protists were observed to phagocytose B. minutum.

Figure 1 Confocal and DIC images of whole Aiptasia egesta.

(A–C) Confocal images of whole Aiptasia egesta. Magenta: Breviolum minutum chlorophyll is detected with excitation at 633 nm and emission at 647–722 nm. Yellow: Cyanobacteria phycobillin (white arrowhead), host tissue (small white arrow), and B. minutum pyrenoids (yellow arrowhead) are detected with excitation at 488 nm and emission at 491–572 nm. Cyan: Pigment from an unknown protist (white arrow) is detected with excitation at 405 nm and emission at 410–470 nm. (D–E) Fired nematocysts (magenta arrows) and empty capsules (black arrowheads) are seen on the superficial surface of egesta under DIC microscopy. Scale bars: (A) 100 µm and (B–E) 20 µm.

Algal physiology

Egesta were released at night, beginning five to six hrs after lights off and peaking at seven to eight hrs after lights off (Fig. 2A). The rate of release slowly decreased until lights were turned on, and release decreased to background levels at the onset of light. The timing of dish cleaning prior to time-lapse imaging had a minor effect on timing of egesta release by delaying the time of peak egesta release by one hr (Fig. 2A). We noticed that the size of egesta was not uniform (data not shown)—presumably containing different amounts of algae, but we were unable to quantify the size of egesta and the number of symbionts per egestum in our experiments. Therefore, the time series data reflects only the number of discrete pellets that were released by anemones and is a proxy for the true number of expelled algae.

Figure 2 The physiology of algae from egesta.

(A) Amount of egesta released per anemone every hour over 24 hours. Experiments 1 and 2 began at 17:00 and experiments 3 and 4 began at 23:00. Black and white bars on x-axis depict dark and light periods, respectively. (B) The mitotic index of algae in egesta and in the host. Dots indicate individual samples and lines connecting dots indicate paired samples. The difference was statistically significant (p = 0.01, Paired t-test). (C) The maximum quantum yield of photosystem II of algae over time as measured by FRRf. Vertical lines indicate standard deviations around the mean. (D) Symbiont density in aposymbiotic Aiptasia inoculated with algae in culture, from egesta, or freshly isolated from hosts. Horizontal lines indicate mean symbiont density. Differences were not statistically significant (p = 0.06, Kruskal-Wallis test).

The released algae were alive and biologically active. The mitotic index of algae from egesta was higher than that of algae in hospite (Fig. 2B; p = 0.01, paired t-test), and the Fv/Fm of algae in egesta, disrupted and intact, at time zero was the same as those in symbiotic anemones (Fig. 2C). Algae from egesta then declined in Fv/Fm over time, with Fv/Fm remaining relatively high for two days before declining rapidly, with chlorophyll fluorescence becoming unmeasurable by day five. Mechanically disrupting the egesta did not have a significant impact on algal photophysiology. Algae from culture started at a lower Fv/Fm than those from egesta or anemones, but Fv/Fm did not decline over time. Cell staining with Evans Blue dye found that 1.5% (17 out of 1,132 total cells) of algae from egesta were dead. Algae from egesta were capable of initiating symbiosis with aposymbiotic Aiptasia, reaching symbiont densities just as high as Aiptasia inoculated with cultured and freshly isolated algae (Fig. 2D). Negative controls remained aposymbiotic for the duration of the experiment (data not shown). While freshly isolated symbionts reached the highest density of symbionts, the difference was not significant (p = 0.06, Kruskal–Wallis test).

Gene expression

Nitrogen transporter and nitrogen metabolism genes, AMT, NRT, NUP, and NR, were significantly upregulated in algae from egesta and algae in hospite compared to culture (Fig. 3). Expression levels of these genes were not significantly different between algae from egesta and algae in hospite. Glutamine synthetase, GT, gene expression showed no significant difference between algal sources. Expression levels of psbA and sugar transporter, SWEET1, were highest in cultured algae compared to algae from egesta and algae in hospite. Overall, with the exception of glutamine synthetase, expression levels of the tested genes were similar between algae from egesta and algae in hospite compared to algae in culture.

Figure 3 Gene expression of nutrient-related, photosynthesis, and sugar transporting genes between cultured algae, algae from egesta, and algae in hospite.

With the exception of GS, Expression of genes was similar in algae from egesta and algae in hospite compared to that of cultured algae. Bars indicate standard deviations around the mean. Transcript IDs are listed under gene abbreviations based on Xiang et al. (2015). Stars indicate statistically significant differences calculated on ΔΔCt values as determined by ANOVA and post-hoc Tukey’s tests. * p < 0.05, ** p > 0.01, *** p < .0001, **** p < 0.00001.

Discussion

Egesta are sticky communities of microbes

Egesta primarily consisted of B. minutum as previously reported (Steele, 1975) and these egesta were released separately from digested food. In addition, egesta contained host material and a variety of microbes (Fig. 1). In sample handling, we noted that egesta were sticky and tended to adhere to plastic surfaces. The stickiness is possibly caused by the presence of fired nematocysts that were present on the egesta surface (Figs. 1D and 1E). The presence of nematocysts, both fired and unfired, were previously described in algae-containing egesta from the sebae anemone, Heteractis crispa (Alan Verde, Cleveland & Lee, 2015). The authors noted that egesta released by H. crispa were consumed by symbiotic anemonefish (Alan Verde, Cleveland & Lee, 2015). It is not known if other animals consume Aiptasia egesta in nature, but if so, they may aid in algal dispersal, as algae often survive digestion (Parker, 1984; Grupstra et al., 2021; Grupstra et al., 2022).

Evidence for the preferential expulsion of dividing algae

The mitotic index of algae was higher in egesta than in hospite, a result that agrees with previous findings in Aiptasia and several corals (Baghdasarian & Muscatine, 2000). It was hypothesized by Baghdasarian & Muscatine (2000) that the host preferentially expels dividing algae. This hypothesis is well-supported from a temporal perspective because the egesta were released at night, and several studies have shown that peak cell division occurs at night in cultured Symbiodiniaceae (Fitt & Trench, 1983; Smith & Muscatine, 1999; Yamashita & Koike, 2016; Fujise et al., 2018). In contrast, algal cell division was not elevated at night in hospite in Aiptasia (Smith & Muscatine, 1999). In that study, however, the dividing algae may have been expelled by the host prior to sampling, leading to the detection of unchanged cell division at night (Smith & Muscatine, 1999). Measuring the mitotic index of algae from egesta that are released over a diurnal cycle may help complete the picture of algal population dynamics in hospite.

Nutrition may also explain the higher division rates in algae from egesta compared to those in hospite. As nutrient availability has been shown to strongly predict cell division rates in Symbiodiniaceae (Smith & Muscatine, 1999; Karako-Lampert et al., 2005; Tivey, Parkinson & Weis, 2020), the host may be preferentially expelling algae that sequester more nutrients for cell division than their more cooperative counterparts that remain undivided in hospite. Indeed, in competitive inoculation experiments, the symbiont Durusdinium trenchii, known to sequester more nitrogen than B. minutum in Aiptasia (Sproles et al., 2020), failed to proliferate in Aiptasia in the presence of the homologous B. minutum (Gabay et al., 2019). The preferential expulsion of dividing algae could be a mechanism for the removal of uncooperative symbionts. Future experiments using methods such as stable isotope analysis and NanoSIMS could investigate whether expelled algae sequester more nutrients than their counterparts in hospite and whether they continue to sequester more nutrients when establishing symbiosis with new hosts.

Timing of algal release varies depending on host taxa

The night-time release of algae in egesta by Aiptasia is not shared by other hosts. In several corals, including Pocillopora damicornis and Acropora digitifera, expulsion rates increased with the onset of light and peaked in the middle of the day (Stimson & Kinzie, 1991; Koike et al., 2007). Other studies in the coral Stylophora pistillata, the soft coral Xenia macrospiculata, and the giant clam Tridacna crocea found no clear diel pattern for algal release (Hoegh-Guldberg, McCloskey & Muscatine, 1987; Umeki et al., 2020). Some corals released algae at night, with peak expulsion rates occuring at night for the corals Millepora dichotoma and Heteroxenis fuscescens (Hoegh-Guldberg, McCloskey & Muscatine, 1987). These contrasting patterns may reflect the actual differences in patterns of algal release between species, or it could reflect differences in sampling methodology. As peak algal motility occurs at midday for Symbiodiniaceae (Fitt & Trench, 1983; Yamashita & Koike, 2016), some sampling approaches could miss subpopulations of algae in the vessel depending on time of sampling.

Algae from egesta are competent symbionts, but are short-lived ex hospite

Algae from egesta were short-lived ex hospite compared to their cultured counterparts, possibly due to nutrient limitation and/or the unknown effects of being in a microbial community (Fig. 2C). This supports the hypothesis that expelled algae do not form stable populations ex hospite (Thornhill et al., 2017). However, freshly expelled algae from egesta had high viability and were fully capable of initiating symbiosis with aposymbiotic hosts (Fig. 2D). In nature, larval and juvenile recruits probably rely on the continuous release of algae from adult hosts for a source of symbionts (Thornhill et al., 2017). This is supported by studies that found that the presence of adult coral colonies, presumably releasing algae, aided in symbiont acquisition by juvenile corals in aquarium experiments (Nitschke, Davy & Ward, 2016; Ali et al., 2019). In addition, a study found that a giant clam harboring Symbiodiniaceae released viable symbionts that were able to establish symbiosis with Acropora tenuis larvae (Umeki et al., 2020). Egesta are also potential vectors for the transmission of the microbiome between adults and recruits. In corals, there is evidence that Acropora tenuis and Pocillopora damicornis expel beneficial microbes into the seawater after spawning (Ceh, Van Keulen & Bourne, 2013). Additional experiments are required to test these hypotheses, particularly in coral recruits.

Evidence that algae from egesta have a similar nutritional status to algae in hospite

In general, in our study, gene expression profiles of algae from egesta were more like those of algae in hospite than those of algae in culture. Several of the tested genes were nitrogen transporters or metabolizers. Transcriptional regulation of these genes is rapid (less than a day) in B. minutum and other marine algae in response to nutrient levels (Hildebrand, 2005; Kang et al., 2007; Xiang et al., 2020). As the algae from sampled egesta were 1–2 days old, changes in transcript levels for these genes should have occurred in this experiment if nutrient levels had significantly changed once algae were expelled by their hosts. Our data suggest that algae from egesta have a similar nutritional status to algae in hospite, challenging the hypothesis that the host is actively limiting nutrient transport to the symbiont (Maor-Landaw, Van Oppen & McFadden, 2020; Xiang et al., 2020; Cui et al., 2022). The identification of nutrient-related genes as differentially expressed in symbiosis studies that used cultured algae to represent the ex hospite state may instead reflect the nutrient-rich state of algae in culture (Maor-Landaw, Van Oppen & McFadden, 2020; Xiang et al., 2020; Cui et al., 2022). In addition, we found that the sugar transporter SWEET1 (s6_35311), was not differentially regulated with symbiosis, challenging the hypothesis that this gene copy of SWEET1 functions to transfer photosynthate from the symbiont to the host (Xiang et al., 2020). This current study, however, did not measure algal growth rates in culture—therefore, we were unable to determine the growth phase of the algae, which can have significant effects on algal physiology and gene expression (Droop, 1975; Mansour, Volkman & Blackburn, 2003; Xiang et al., 2020). Nevertheless, we found that algae in culture had the lowest expression rates of nutrient transporters and nutrient metabolizers, indicating a nutrient-replete phenotype compared to algae from egesta and algae in hospite. Altogether, these results warrant further testing of gene expression in algae from egesta using high throughput methods to develop a better understanding of symbiosis.

Practical considerations for the use of Aiptasia egesta in experiments

Based on findings in this study, we argue that it is critical to further explore the biology of expelled algae as a representation of the ex hospite state of Symbiodiniaceae to better understand cnidarian-algal symbiosis. To facilitate their use by researchers, we discuss below practical considerations for the use of Aiptasia egesta in experiments.

We found that while egesta are primarily composed of Symbiodiniacean algae, they are also populated with microbes, including several protists and cyanobacteria. In our experience, processing of egesta by homogenization and separation by density centrifugation yielded samples of high purity. Collecting ample biological material for use in qPCR and inoculation experiments was difficult. To obtain 1 × 106 algal cells from egesta, pellets had to be collected six times every other day from approximately twenty anemones with an average oral disk size of 5.4 mm. We also found that collection was variable between days and yield was not consistent (data not shown). For collection of egesta, if Aiptasia are housed without water flow, we recommend using a glass pasteur pipet, as the egesta tended to stick to plastic pipets. Egesta can also be mistaken for pedal lacerates, so it is recommended that a dissection microscope be used to aid in identification. It is possible that various factors, such as anemone size, light levels, and feeding, all influence the rate of egesta release, but these variables were not empirically tested. Another consideration is the induction of algal release by mild stress to increase sample biomass, however, the health of the algae may be affected, so this method will require further experimentation.

Conclusions

Overall, this study establishes the foundation for using Aiptasia egesta in experiments as a representation of algae in the ex hospite state. We found that egesta were released at night, and that algae from egesta had a higher mitotic index than algae in hospite, were photosynthetically active but short lived, and could establish symbiosis with new hosts. Finally, we found that algae from egesta had a similar gene expression profile to that of algae in hospite compared to algae in culture, warranting future high-throughput studies to determine symbiosis-specific genes in this comparison. Although there are limitations to the use of algae from egesta, their use in experiments alongside algae in culture and algae in hospite can provide valuable insight into cnidarian-algal symbiosis.

Supplemental Information

Supplemental Information 1 Primer sequences used in qPCR

Individual primers including primer efficiencies for each gene tested.

Click here for additional data file.

Data S1 Raw data

Raw data including symbiont density, mitotic index, qPCR, egesta release time-lapse data, and Fv/Fm data.

Click here for additional data file.

We would like to thank animal care staff, Bell Hansen, Alex Bridge, Lily Miksell, Julia Johnston, Kyra Lenderman, and Noah Tjandra for maintaining Aiptasia used in experiments. We also thank Weis lab members, Erick White, Jun Cai, Keyla Plichon, Dr. Samuel Bedgood and visiting researcher Dr. Lucia Pita for feedback on drafts. We thank Dr. Tingting Xiang at University of North Carolina, Charlotte, for providing Aiptasia animals and algal culture used in experiments. Finally, we would like to thank the three reviewers for their critical feedback that allowed us to improve the manuscript. The members of the Students of Oregon State University’s Z362 Spring 2021 group are Dawson Loehner, Joseph Stewart, Bree Feliciano George, Emily Denise Miller, Sinaiah Sorcha Harrington, Julia Amber Kennedy, Lucas Parvin, Violet Carrillo, Alex Michael Bridge, Kaitlyn Ashley Allison, Kennedy Judith Grant, Jess Alt, Margaret Mae Zackery, Conor Ringwald, Ian Reed, Neosha Hubbs, Madalyn Rose Machinski, Mahala Grace Gilbert, and Bell Hansen.

Additional Information and Declarations

Competing Interests

Author Contributions

Data Availability

The authors declare there are no competing interests.

Shumpei Maruyama conceived and designed the experiments, performed the experiments, analyzed the data, prepared figures and/or tables, authored or reviewed drafts of the article, and approved the final draft.

Julia R. Unsworth performed the experiments, analyzed the data, authored or reviewed drafts of the article, and approved the final draft.

Valeri Sawiccy conceived and designed the experiments, performed the experiments, analyzed the data, prepared figures and/or tables, authored or reviewed drafts of the article, and approved the final draft.

Virginia M. Weis conceived and designed the experiments, authored or reviewed drafts of the article, and approved the final draft.

The following information was supplied regarding data availability:

The raw data for egesta release timing, mitotic index, Fv/Fm, symbiont densities, and qPCR are available in the Supplemental Files.

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
