# Peer review of "Algae from Aiptasia egesta are robust representations of Symbiodiniaceae in the free-living state"

_PeerJ, doi:10.7717/peerj.13796_

## Round 0.1 · original submission · Major Revisions

Dear Shumpei and co-authors,

I have received three contrasting reviews of your study. I'd like to thank the reviewers for their time and efforts in reviewing this work. While all reviewers recognised the scientific value of your work, they have collectively raised a number of issues that will need to be addressed in your revised manuscript. In particular, the scope and data interpretation has been questioned by several reviewers, and it appears that the manuscript writing and structure needs considerable improvement.

Overall, the reviewers have provided you with excellent suggestions on how to improve the manuscript, and I be looking forward to receiving your revised manuscript along with a point-by-point response to their comments.

With warm regards,
Xavier

·

Basic reporting

To be honest, the purpose of this study was not clearly understood. Based on the assumption that the origin of symbiotic algae acquired by animals is not a free-living population in the environment, but that they are expelled from other animals, the authors analyzed physiology and gene expression of the symbiotic algae in egesta of Aiptasia in detail, and based on the results, they seemed to propose using the symbiotic algae in the egesta as a standard host-symbiosis study. What would be that study? Given that Aiptasia acquires only a specific Symbiodiniaceae (Breviolum), it may be suggested that the results of this study be used as an experimental standard in the acquisition of symbiotic algae by Aiptasia, but this is not clearly stated. In view of various reports that symbiotic algae become a flagellated swimming stage in a free-living state, and that this swimming ability is necessary for attraction to corals, the results of this study, which did not observe whether they change to a swimming stage, do not seem to be applicable to coral research. So, if the authors are going to propose this as a model for how animals acquire symbiotic algae, they need to limit it to Aiptasia first.
I, too, do not believe that symbiotic algae live solitary in the oligotrophic reef waters, and, like the authors, I believe that expelled symbionts from other animals are the symbiont source for the next animals. However, some researchers, for example Littleman et al. (2008), say that “Free-living Symbiodiniaceae” is a symbiotic source, and we don't know the truth yet, so I think Line 62 is saying too much.
So, my suggestion is to rewrite the introduction as to focus on Aiptasia-symbionts only, not to broaden the story as to cover general of animals including corals, and to incorporate various aspects for the origin of symbiont sources in coral reef environment. Also, sentences describing inappropriateness of using culture strains (line 84) are unclear. There are numbers of studies using culture strains isolated from certain hosts for symbiosis establishment test on the same host (e.g. Umeki et al 2020). If the description was for Aiptasia study only, please mention so.

Experimental design

Line 126. It seems the tested animal was initially symbiosed with a culture strain of Breviolum. Isn’t this conflicting with the argument from Line 80 saying culture strain is not representative of what found in actual hosts?

Line 129 Was artificial seawater used to prepare culture medium F/2? If so, there are many reports arguing most microalgae are not maintainable in an artificial seawater-based medium, means that physiological results of this study were under abnormally unfavorable condition for the culture strain. It might be just a filtered seawater (FSW), wasn’t it?

Line 193 “Dark-adapted photosynthetic efficiency” should be “maximum quantum yield of the photosystem II”

Line 217. This seems to be common to many infection experiments, but is the cell density to be inoculated too high? It's a personal feeling, but if you give them so much symbiotic algae, they'll probably accidentally take it in, and do they really represent what's going on in the field? This is fine this time, but I recommend testing at a lower density next time to highlight egesta's contribution to symbiosis formation.

Validity of the findings

I did not understand the purpose of the description of gene expression in culture and in the host (from line 93). Are the authors trying to tell us the difference between the nutrient environment in egesta and a culture that is saturated with various nutrients? Or are they introducing AMT, NRT, NR, etc., to emphasize that the enzymes involved in nutrient uptake are not indicative, and that SWEET1 alone is a good indicator of the difference between conditions in the host and conditions away from the host? This fuzzy feeling is felt similarly in the text from Line 414 of the discussion. Genes for nutrient uptake remained the same in egesta and host and are much higher expressed than those in culture. This seems to indicate that the symbiotic state is not as present in the hypertrophic environment as in culture. If so, why the authors is “Challenging the hypothesis that the host is active limiting nutrient transport to the symbionts”? The implications of L 428 for high SWEET 1 expression are also unclear. Do they want to argue that this gene is not upregulated by host as suggested by Xiang et al 2020, and this is what they want to emphasize in this study?

Additional comments

Result
Fig. 1 No black arrows in D or E. Any specific reason for using 633 nm for chlorophyll excitation, not the blue one?

Line 308. Numbers of the expelled egesta were shown. However, we do not know the regular size of egesta, size variation, and average symbionts density in an egesta, which make us difficult to imagine whether the symbiont releases were large or small.

Line 315 “Photosynthetic health”. FV/Fm is just a quantum yield of the most PSII upstream and does not indicate the healthiness of overall photosynthesis.

Discussion
Line 394. This argument is highly agreeable, but probably limited to giant clams (see Umeki) and Aiptasia. In our experiments, the symbiotic algae released by corals died within 24 hours. In both Aiptasia and giant clams, the expelled symbiotic alga remain active (in egesta or feces) for a relatively long time, which may be advantageous to supply symbiotic source for others.

Reviewer 2 ·

Basic reporting

I enjoyed reading the submission by Maruyama and colleagues. Overall, this ms adheres to PeerJ professional and structural guidelines. My three primary concerns with regards to basic reporting are described below.

1. The taxonomic nomenclature of the focal study species within the submission reflects outdated taxonomy. Replace “Aiptasia” with “Exaiptasia” throughout and cite Grajales & Rodriguez 2014 Zootaxa https://www.mapress.com/zootaxa/2014/f/z03826p100f.pdf .
2. I often felt that the flow of the results and discussion were not tight enough and could be improved. Within these sections there were several brief (2-3 sentence) paragraphs (L313, L322, L394, L437). Flushing out the transitions in and out of these segments ought to help make the story more succinct.
3. Information on the ex situ culture experiment need to be described in more detail (see comment on L263 of methods). Cell densities from the duration of the culture experiment do not appear to be reported in the supplement

Experimental design

The clever and creative experimental design are among the submission’s major strengths. The ms does an excellent job detailing the importance of the knowledge gap and difficulties associated with studying environmental Symbiodiniaceae populations. As such, using Symbiodiniaceae from anemone egesta as a proxy for studying the photophysiology, cell cycle, and gene expression of environmental conterparts is creative. I imagine this will inspire future research in the study system. I also wish to commend the authors on their successful incorporation of undergraduate research-based teaching and inclusion course members as co-authors.

Validity of the findings

The ms describes the proper controls and provides most necessary raw data. For the most part, the conclusions are justified by the results at hand (point by point comments below).

Additional comments

Thank you, Maruyama and colleagues, for your interesting and creative submission to PeerJ! My point by point comments are below.

Abstract
L31-32: omit from abstract to improve brevity
L33: define egesta

Introduction
L58-63: I think these statements over-speculate about the transient nature of environmental populations of Symbiodiniaceae. I also recommend citing Grupstra et al 2021 Animal microbiome which documents an abundance of viable Cladocopium and Durusdinium in the feces of corallivorous fishes. Some additional papers about environmental Symbiodiniaceae populations are also listen in my comments on the discussion.
L91-93: two sentences in a row are started with the phrase “using algae.” Consider revising
L96-97: Add the citations for the data papers evaluating nutrient availability, gene expression, and cell division rates
L121: finish off this paragraph with an additional sentence tying together how this paper demonstrates that expelled algae are an effective proxy for studying environmental populations of Symbiodiniaceae,

Methods
L138: how frequently were the anemones checked for egesta released?
L237: is there a reference for the Cyclophilin housekeeping gene?
L263: Major concern -- more detail is needed on the details of harvesting samples from the ex hospite culturing experiments. The ms needs to verify the growth stage of each flask to ensure that samples were generated during comparable physiological states (ideally exponential growth). Here are some examples of papers demonstrating the physiological & transcriptomic variability of microalgal in between growth stages: Droop 1975 “The nutrient status of algal cells in batch culture”; Mansour et al 2003 “The effect of growth phase on the lipid class, fatty acid and sterol composition in the marine dinoflagellate, Gymnodinium sp. in batch culture”; Yoon et al 2002 “Combined transcriptome and proteome analysis of Escherichia coli during high cell density culture.”

Results
L311: is this data included in the supplement?
Paragraph at L328: include results from output of statistical analyses.

Discussion
L366: cite the original data papers evaluating nutrient availability and cell division rates
L370: swap “cooperative” for “homologous”
L373: I think this statement also over-reaches with speculation. Consider revising or removing
L375-377: I agree! I also think a benefit of the Exaiptasia-Symbiodiniaceae system is its capacity to work with several genera (and therefore species) of Symbiodiniaceae. Consider suggesting comparative approaches along these lines.
Paragraph at L379: this could also be due to elevated rates of stony coral feeding at night?
L394: is the FSW sterile? Perhaps bacterial symbionts are missing
L397: I think this statement overstates the conclusions and as it challenges our understanding of Symbiodiniaceae population genetics. For example, if the same genotype of D. trenchii is dominant throughout the Caribbean, how was its rapid, invasive population spread possible given the stated hypothesis about the transient nature of environmental populations (i.e., Pettay et al 2015 PNAS)? Here are some papers reporting population dynamics of Symbiodiniaceae in the water column: Decelle et al 2020 (https://www.sciencedirect.com/science/article/pii/S0960982218312193) and Raina et al 2022 (extended data 1: https://www.nature.com/articles/s41586-022-04614-3)
Paragraph at L413: I am surprised that the utility of proteomes in phenotyping (a la Maryuma & Weis 2021) is not mentioned here.
L422: Nice! I agree!
L441-442: I think this statement articulates why this submission represents a considerable breakthrough in the field. Consider alluding to earlier on in the text.
L442: replace col2lected with collected
L450-452: would perturbing anemone oral disks with a glass Pasteur pipette induce egesta release (without adding confounding variables)?

Reviewer 3 ·

Basic reporting

The manuscript “Algae from Aiptasia egesta are competent symbionts and more closely represent the free-living state of Symbiodiniaceae in nature than do algae in culture” by Maruyama et al. explores the physiology of aiptasia egesta and compares it to an algal culture. I commend the authors for doing this work. However, the overall manuscript needs significant revision and streamlining of the results and discussion. This study could be interesting for the aiptasia community, but its current form should be revised.

More broad concerns/critiques
1. I felt the manuscript could have been written as a description of the biology of egesta. The authors should have streamlined the claims and only discussed what their data tells us.

2. The overall justification and rationale of the study are somehow weak, and there are points in the discussion that don’t align with it.

Experimental design

Methods:
L207: Please mention and specify the light intensity used.

Did the authors use negative control (seawater only) for the inoculation experiment? This should be included in the method section so we can tell that the four-day exposure to L:D was not the reason we are seeing algal signals in aposymbiotic aiptasia.

Primer design. The authors used Cyclophilin as a housekeeping gene. This should be justified. For example, show expression that the gene does not change in expression over time or cite a study that used the same gene.

Validity of the findings

Results

The authors reported no significant difference between egesta and in hospite gene expression using seven genes. How can we tell that the signal was not confounded by non-intact/dead cells incorporated in the extracted egesta?
A supplemental figure showing the ratio of intact and non-intact cells could be added to the supplementary materials.

Additional comments

Discussion

In Line 340-341, the authors mentioned about a variety of microbes. However, the current methods employed in the study to support this is not sufficient. More specifically, microscopy resolution compared to 16S sequencing is very different.

In Line 364-365, the authors mentioned that a higher mitotic index of egesta than in hospite could be due to nutrient availability. This statement doesn’t align with the justification above that egesta can be a representation of algal symbiont lifestyle than that of the cultured algae.

In lines 394-397, the authors mentioned that egesta is short-lived, which could be a limitation if we suggest its use for comparative study instead of algal culture. It’s also challenging to employ this in coral research when we have to wait at nighttime to have enough expelled symbionts samples.

Line 402-404 is speculative, and this statement should be removed unless the authors can cite a study. The results from this study don’t support this.

The whole section of 412-413 used select genes, and the statement that “egesta to represent the ex hospite state can remove nutrition as confounding variable” could not strongly support this claim.

---

## Round 0.2 · Minor Revisions

Dear Dr. Maruyama and co-authors,

Please address the remaining minor issues highlighted by the reviewers on your revised manuscript.

I will be looking forward to receiving your final version of the manuscript along with a point-by-point response to their comments.

With warm regards,
Xavier

·

Basic reporting

I still wonder whether the symbiont of egesta signifies one of their life stages, i.e., a free-living state. It seems to me that the symbionts in the egesta were simply dragging the environment within the host. In addition, it is difficult for me to imagine that the cellular environment in the host cells is oligotrophic even if the host animals live in the oligotrophic seawater, and therefore, the logical development that the nutrient-rich culture environment does not represent the actual situation is uncertain. However, it is valuable in the sense that the symbionts kept photosynthetic activity for long time of 48 hours even if they left from the host, and in addition, the current observation gives information to the hot discussion on what Symbiodiniaceae as a symbiotic source of the coral derives from.

Experimental design

The authors provide new information on the matters pointed out previously, and the result of live or death discrimination by Evans Blue is also interesting. The low rate of dead cells was surprising.

Validity of the findings

The observation gives another important information on how animals including corals acquire symbiotic algae and what they are derived from, and is worthy of publication.

Reviewer 2 ·

Basic reporting

see additional comments

Experimental design

see additional comments

Validity of the findings

see additional comments

Additional comments

I thank the authors for their efforts restructuring the ms. I believe it is an exciting contribution to the fields of symbiosis biology and coral reef ecology. My remaining comments are minor and primarily responding to the rebuttal.

1. Photosynthetic health metrics. I agree with reviewer 1, Fv/Fm indicates PSII photochemical efficiency, which is important for phenotyping, but is not suitable as a sole indicator of photosynthetic health. I encourage the ms to reconsider revising their use of the phrase “photosynthetic health” to more accurately depict the contents of their data and embrace the nuances associated with photobiology (i.e., something along the lines of ‘algal photophysiology’). The utility of Fv/Fm and chla fluorescence are further described in Suggett et al 2009 MEPS https://doi.org/10.3354/meps07830 and Kalaji et al 2017 Photosynthetic Research https://doi.org/10.1007/s11120-016-0318-y
2. Anemone taxonomy. I respectfully disagree with the rebuttal on Exaiptasia taxonomy. Many papers include both the updated taxonomy and informal name (see Dungan et al 2020 Symbiosis; Strumpen et al 2022 symbiosis; Siddiqui et al 2015 Aquatic Toxicology). I see this as a critical component to basic reporting, promoting reproducible research, and avoiding confusion for new students/researchers. I appreciate that the formal binomial was added to the abstract and introduction and request that it be added to the first mention in the methods (L125)
3. Algal culturing methods. Typically, cell growth calculations are displayed as maximum growth rate (mu), but here are reported as cell yield. The ms now reports the age at which cultures were harvested (L370). The strain of B. minutum could be in either exponential or stationary phase by day 21 – this distinction would greatly impacts its physiology + nutritional status and the broader conclusions of the ms. I recognized it is likely a) not feasible to run additional experiments and b) additional experiments would induce batch effects in qpcr data; but request that this caveat is incorporated in the discussion sections at L422 & L442 (see references from first round of review).
4. Reviewer 1 expressed concern with regards to usage of artificial seawater (ASW) in algal cultural media and potential for unintentionally inflicting stress on the microalgae. I agree with the author’s response about ASW as an effective culturing option. The nutrient concentrations in common algal culture recipes are replete (see Krueger 2020 https://www.researchgate.net/publication/339788469_Media_recipes_commonly_used_for_Symbiodiniaceae_cultivation). As such, ASW culturing is unlikely to induce treatment artifacts.


L359: change “sebae” to “sea”

---

## Round 0.3 · accepted · Accept

Dear Dr Maruyama and co-authors,

I am pleased to accept your revised manuscript for publication in PeerJ. I'd like to thank the reviewers for their time and thank you for this valuable scientific contribution.

With warm regards,
Xavier